# Identification of Thiazolo[5,4-*b*]pyridine Derivatives as c-KIT Inhibitors for Overcoming Imatinib Resistance

**DOI:** 10.3390/cancers15010143

**Published:** 2022-12-26

**Authors:** Yunju Nam, Chan Kim, Junghee Han, SeongShick Ryu, Hanna Cho, Chiman Song, Nam Doo Kim, Namkyoung Kim, Taebo Sim

**Affiliations:** 1KU-KIST Graduate School of Converging Science and Technology, Korea University, 145 Anam-ro, Seongbuk-gu, Seoul 02841, Republic of Korea; 2Severance Biomedical Science Institute, Graduate School of Medical Science, Yonsei University College of Medicine, 50 Yonsei-ro, Seodaemun-gu, Seoul 03722, Republic of Korea; 3Chemical Kinomics Research Center, Korea Institute of Science and Technology, 5 Hwarangro 14-gil, Seongbuk-gu, Seoul 02792, Republic of Korea; 4Voronoibio Inc., 32 Songdogwahak-ro, Yeonsu-gu, Incheon 21984, Republic of Korea

**Keywords:** c-KIT, GIST, GIST-T1, HMC1.2, imatinib resistance, thiazolo[5,4-*b*]pyridine

## Abstract

**Simple Summary:**

c-KIT has been regarded as a promising therapeutic target against gastrointestinal stromal tumor (GIST). Overcoming drug resistance of c-KIT inhibitors including imatinib is required. We designed and synthesized novel thiazolo[5,4-*b*]pyridine derivatives and performed structure-activity relationship (SAR) studies to overcome imatinib resistance. The SAR studies led to the identification of the derivative **6r** as a potent c-KIT inhibitor. The derivative **6r** is capable of strongly inhibiting a c-KIT V560G/D816V double mutant that is resistant to imatinib and remarkably attenuates proliferation of GIST-T1 and HMC1.2 cancer cells. Moreover, **6r** possesses differential cytotoxicity on c-KIT D816V Ba/F3 cells relative to parental Ba/F3 cells. Kinase panel profiling revealed that **6r** has reasonable kinase selectivity. Furthermore, **6r** not only blocks migration and invasion, but also suppresses anchorage-independent growth of GIST-T1 cells.

**Abstract:**

c-KIT is a promising therapeutic target against gastrointestinal stromal tumor (GIST). In order to identify novel c-KIT inhibitors capable of overcoming imatinib resistance, we synthesized 31 novel thiazolo[5,4-*b*]pyridine derivatives and performed SAR studies. We observed that, among these substances, **6r** is capable of inhibiting significantly c-KIT and suppressing substantially proliferation of GIST-T1 cancer cells. It is of note that **6r** is potent against a c-KIT V560G/D816V double mutant resistant to imatinib. Compared with sunitinib, **6r** possesses higher differential cytotoxicity on c-KIT D816V Ba/F3 cells relative to parental Ba/F3 cells. In addition, kinase panel profiling reveals that **6r** has reasonable kinase selectivity. It was found that **6r** remarkably attenuates proliferation of cancer cells via blockade of c-KIT downstream signaling, and induction of apoptosis and cell cycle arrest. Furthermore, **6r** notably suppresses migration and invasion, as well as anchorage-independent growth of GIST-T1 cells. This study provides useful SAR information for the design of novel c-KIT inhibitors overcoming imatinib-resistance.

## 1. Introduction

c-KIT, a class III receptor tyrosine kinase, consists of five immunoglobulin-like domains, one transmembrane domain, one juxta-membrane domain, and two split kinase domains [1]. Binding of a stem cell factor (SCF) ligand to c-KIT induces the c-KIT dimerization, leading to subsequent activation of its intracellular signaling cascades [2]. Dysfunctions of c-KIT, such as gain of function caused by overexpression and point mutation, result in c-KIT activation and tumorigenesis [3,4]. Activation of c-KIT is observed in various cancers including GIST, mast cell tumors, and malignant melanomas [4,5,6]. In particular, c-KIT activation is deeply associated with GISTs which are the most common gastrointestinal tumor [7,8]. More than 90% of 419 GIST cases are associated with c-KIT activation and 6−7% of the cases are implicated with PDGFRα mutations [9].

Mutations within exon 11 encoding the juxta-membrane domain of c-KIT, with a relative frequency of 67%, are most common in GIST [10]. The mutations in exon 9 (IgG-like D5 domain of c-KIT) of which the majority are duplications of A502_Y503 occur at nearly 15% in GIST [11,12]. Notably, secondary c-KIT mutations causing imatinib resistance have been known to occur most commonly in exon 13/14 (ATP-binding domain of c-KIT) and exon 17/18 (activation loop of c-KIT) [13,14,15]. The c-KIT T670I gatekeeper mutant within exon 13 corresponds to the BCR-ABL T315I gatekeeper mutant and causes drug resistance through constitutive activation and steric hindrance [16,17]. D816H/V mutants occurring in the activation loop of c-KIT are capable of accelerating auto-activation rather than wild-type c-KIT, which leads to resistant to imatinib [18]. Moreover, the double mutant of c-KIT V560G/D816V has disruption of auto-inhibitory mechanism and SCF-independent constitutive activation of c-KIT [19].

Imatinib, an innovative BCR-ABL kinase inhibitor against chronic myelogenous leukemia, has been approved as a first-line treatment for advanced GIST (Figure 1) [20]. However, 10–20% of patients taking imatinib show secondary mutations that decrease c-KIT sensitivity to imatinib [21]. Sunitinib, a second-line therapy for the treatment of GIST, possesses inhibitory activity against c-KIT V654A and gatekeeper mutant c-KIT T670I [13]. However, preclinical studies have revealed that both imatinib and sunitinib do not effectively inhibit exon 17 *KIT* mutations (activation loop mutants) [14]. Regorafenib was approved for the third-line treatment against imatinib-and sunitinib-resistant GISTs but it has moderate activities against the secondary mutations [22,23]. The potent c-KIT/PDGFRα inhibitor avapritinib has been approved for the GIST treatment [24]. However, secondary PDGFRα mutations (Val658Ala, Asn659Lys, Tyr676Cys, and Gly680Arg) have been observed in patients with drug resistant recurrent GIST [25]. Ripretinib, the fourth-line treatment approved for GIST, interacts reversibly to both the switch pocket and activation loop of c-KIT [26]. Although ripretinib is capable of inhibiting a wide range of c-KIT and PDGFRα mutants, it shows an overall response rate of 9.4%, suggesting additional resistance and disease progression [27].

In addition to clinically approved drugs, various scaffolds have been used as c-KIT inhibitors. A potent dual c-KIT/PDGFRα 1*H*-pyrazolo[3,4-*b*]pyridine inhibitor for the treatment of imatinib-resistant GISTs has been reported [28], but its anti-proliferative activity against TEL-c-KIT D816V Ba/F3 cells was not significant (GI_50_ > 10 μM). The derivatives having the 2-aminothiazole scaffold switch off activated c-KIT to its inactivated state, and possess activities against constitutively activated mutants in the activation loop [29]. Moreover, the thiazolo[5,4-*b*]pyridine scaffold has been used to discover various kinase inhibitors against PI3K [30], ITK [31], BCR-ABL [32], RAF [33], and VEGFR2 [34]. Thiazolo[5,4-*b*]pyridine derivatives have distinct types of binding modes depending on target kinases. The 4-nitrogen of thiazolo[5,4-*b*]pyridine is hinge-binding motif of PI3K kinase inhibitors while the 1-nitrogen and 2-amino group of the scaffold form H-bonding contact with the ITK kinase hinge region. In addition, the 5-position of thiazolo[5,4-*b*]pyridine has been functionalized to target the ATP-binding site of BCR-ABL, RAF, and VEGFR2. It is worthwhile to note that this study has, for the first time, reports functionalization on the 6-position of thiazolo[5,4-*b*]pyridine scaffold to identify novel c-KIT inhibitors.

In an effort to overcome drug resistance, we synthesized 31 novel thiazolo[5,4-*b*]pyridine derivatives and carried out a structure-activity relationship study against c-KIT enzyme and c-KIT-activated cells. The results of the SAR study for thiazolo[5,4-*b*]pyridine derivatives revealed that **6r** possesses higher enzymatic and anti-proliferative activities than imatinib and comparable activities relative to sunitinib. Notably, **6r** has 8.0-fold higher enzymatic inhibitory activity (IC_50_ = 4.77 μM) against c-KIT V560G/D816V double mutant and 23.6-fold higher anti-proliferative activity (GI_50_ = 1.15 μM) on HMC1.2 cells harboring both V560G and D816V c-KIT mutations compared with imatinib. Moreover, **6r** exhibits a reasonable selectivity in the biochemical kinase panel profiling. Furthermore, **6r** suppresses proliferation of cancer cells (GIST-T1 and HMC1.2) and c-KIT D816V Ba/F3 cells via blockade of c-KIT downstream signaling, and induction of apoptosis and cell cycle arrest. Notably, **6r** remarkably blocks migration and invasion, as well as anchorage-independent growth of GIST-T1 cells.

## 2. Materials and Methods

### 2.1. Chemistry

All commercial reagents and solvents were purchased from chemical suppliers and directly used for synthesis without purification. Reaction monitoring was performed by thin-layer chromatography (TLC) analysis using a UV lamp, ninhydrin, or *p*-anisaldehyde stain for the compound detection. Reaction products were purified by column chromatography on silica gel (230−400 mesh). Purities of all compounds were analyzed by Waters LC/MS system and shown to be over 95%. ^1^H and ^13^C NMR spectra were recorded either by a Bruker 300 MHz FT-NMR (300 MHz for ^1^H, and 75 MHz for ^13^C) or a 400 MHz FT-NMR (400 MHz for ^1^H, and 100 MHz for ^13^C) spectrometer. The synthetic procedures and chemical characterizations of all compounds are described in Appendix A.

### 2.2. Cell Culture and Reagent

GIST-T1 (Cosmobio, Tokyo, Japan, # PMC-GIST01C-COS) cells were cultured in DMEM (Welgene, # LM001-05) and HMC1.2 (Merck, Darmstadt, Germany, # SCC062) cells were cultured in IMDM (Welgene, # LM004-01). Parental Ba/F3 (DSMZ, Braunschweig, Germany, # ACC300) cells were cultured in RPMI (Welgene, # LM011-51) with 10 ng IL-3/mL (Enzo, # ALX-201-821). The culture media was supplemented with 10% fetal bovine serum and 1% penicillin/streptomycin solution. The cells were maintained in a humidified atmosphere containing 5% CO_2_ at 37 °C.

### 2.3. In Vitro Kinase Assay

Biochemical kinase assays on c-KIT protein kinase were performed at Reaction Biology Corp. (San Diego, CA, USA). Compounds were tested with 10 μM ATP in a 10-dose IC_50_ mode with 3-fold serial dilution. The bioluminescent-based kinase assay was tested using an ADP-Glo assay kit. c-KIT (V560G/D816V) kinase was purchased from Promega (# VA7063). Compounds were tested with 10 μM ATP in a 10-dose IC_50_ mode with 3-fold serial dilution. The kinase assay followed the manufacturer’s instruction. IC_50_ values were calculated by GraphPad prism 8.0 software (GraphPad Software, San Diego, CA, USA).

### 2.4. Cell Viability Assay

Cells (GIST-T1: 5.0 × 10^3^; HMC1.2: 1 × 10^4^) were seeded in 96-well plates. Each compound was treated in each well at 10 dose points of 3-fold serial dilution. After treatment with each compound for 72 h, CellTiter-Glo solution (Promega, # G7572) was added to each well. Anti-proliferation activity was measured the luminescence using a 96-well plate reader (EnVision 2013).

### 2.5. Molecular Docking Study

The X-ray co-crystal structure of c-KIT complexed with imatinib (PDB code: 1T46) was retrieved from the Protein Data Bank and loaded into Maestro software (Schrödinger Release 2020-4, New York, NY, USA). The Protein Preparation Wizard was executed for replenishment of missing residue, addition of hydrogens, and assignment of bond orders. Restrained energy minimization was applicated in the OPLS3e force field. Ligand **6r** was prepared using the LigPrep and the receptor c-KIT grid was generated considering the imatinib binding pocket. Ligand docking of **6r** on c-KIT was performed using GLIDE.

### 2.6. Kinase Panel Profiling

Kinase panel profiling was conducted by Reaction Biology Corp. (San Diego, CA, USA). A single dose (1.0 μM) of **6r** was tested against 371 recombinant human kinases in the presence of 10 μM ATP.

### 2.7. Western Blot Analysis

Cells were harvested and lysed using RIPA buffer containing 50 mM HEPES (pH 7.4), 1% Triton X-100, 2 mM EDTA, 150 mM NaCl, 2.5 mM NaF, 5 mM Na_3_VO_4_, and protease inhibitor cocktail tablet (Roche, # 11–878-580–001). Proteins were separated and transferred to an NC membrane. The membranes were blocked using 5% skim milk in TBS-T buffer. The p-c-KIT (Tyr703, # 3073), p-AKT (Ser473, # 9271), p-ERK (Thr202/Tyr204, # 8544), ERK (# 4695), p-PLCγ (Tyr783, # 14008), p-STAT3 (Tyr705, # 4074), and PARP (# 9542) antibodies were obtained from Cell Signaling Technologies. The β-actin (# sc-47778) and STAT3 (# sc-482) antibodies were obtained from Santa Cruz Biotechnology. AKT (A18120) antibody was purchased from ABclonal. Each primary antibody was incubated overnight at 4 °C. The secondary antibody (GenDEPOT) was incubated for 1 h at room temperature. Proteins were detected using ECL substrate.

### 2.8. Cell Cycle Arrest and Apoptosis Analysis

After compound treatment in GIST-T1 and HMC1.2 cells for 24 h, cells were fixed with 70% cold ethanol. Then, cells were incubated with 50 μg propidium iodide/mL solution (PI, Sigma) containing 1 mg RNase A/mL for 30 min in the dark. Cell cycle arrest was analyzed (BD Accuri), and data were processed using BD Accuri^TM^ C6 software. For apoptosis analysis, compound treatment for 24–48 h, cells were harvested and stained with annexinV and PI solution. 1 × 10^5^ cells were analyzed using flow cytometric analysis.

### 2.9. Migration Assay

GIST-T1 cells were seeded in 6-well plates at 2 × 10^6^ cells. Next day, cells were washed with PBS and wounded in the center of each well. Then, GIST-T1 cells were washed with PBS and added to fresh DMEM. After 36 h, the scratch recovery of GIST-T1 cells in each compound-treated groups were observed and taken with a Nikon microscope at a magnification of 1000×.

### 2.10. Invasion Assay

The cell invasion assay was performed using Transwell membrane filter inserts (Corning Costar Corp., Corning, NY, USA). GIST-T1 cells were then suspended in serum-free DMEM at 2 × 10^6^ cells and seeded in the upper chambers. In the lower chambers, the DMEM medium containing 10% FBS was added. After 48 h, the cells migrating down the membrane were stained with crystal violet.

### 2.11. Soft Agar Assay

Anchorage-independent growth was assessed by determining colony formation on soft agar. GIST-T1 cells in DMEM media containing 0.7% agar were plated at 5,000 cells per well. Cells were treated with each compound for 3 weeks, and the media were changed every 3 days. Colonies were stained using iodonitrotetrazolium chloride (Sigma Aldrich). The average number colonies were counted using ImageJ software (version 1.53e, National Institutes of Health, Bethesda, MD, USA)

### 2.12. Statistical Analysis

Statistical analysis was performed using GraphPad Prism 8.0. Graphpad software, San Diego, CA, USA. All data are reported as average ± standard deviation (S.D.). *p* < 0.05 was considered to be statistically significant.

## 3. Results and Discussion

### 3.1. Synthesis of Thiazolo[5,4-b]pyridine Derivatives

The synthetic route for the thiazolo[5,4-*b*]pyridine derivatives is described in Figure 1. To synthesize thiazolo[5,4-*b*]pyridine scaffold, we carried out the aminothiazole formation using commercially available 3-amino-5-bromo-2-chloropyridine **1** and potassium thiocyanate in 75% yield. The amino group of **2** underwent Boc protection to afford compound **3** in 90% yield. Then, **3** was coupled with 2-methyl-5-nitrophenylboronic acid pinacol ester by Suzuki cross-coupling reaction using Pd(dppf)Cl_2_ as catalyst to form **4** in 70% yield. Reduction of the nitro group of **4** afforded the key intermediate **5** in 80% yield. The amide formation of aniline **5** with various carboxylic acids, followed by subsequent Boc deprotection yielded **6a–i** and **6k–w**. Urea formation between aniline **5** and 3-(trifluoromethyl)phenyl isocyanate was followed by subsequent Boc deprotection gave **6j**. The primary amino group of selected derivatives was transformed into the corresponding amides **7a–h** in 21–38% yields.

### 3.2. Structure-Activity Relationships

Enzymatic inhibitory activities of all synthesized thiazolo[5,4-*b*]pyridine derivatives against c-KIT were measured using radiometric biochemical kinase assays (Table 1). We first explored the R_1_ group of the thiazolo[5,4-*b*]pyridine scaffold. The results show that only a 3-(trifluoromethyl)phenyl group (**6h**) resulted in a moderate enzymatic inhibitory activity (IC_50_ = 9.87 μM) among **6a**–**j** possessing five-membered and six-membered aromatic rings. Molecular docking study revealed that the 3-trifluoromethyl group fits well into the hydrophobic binding pocket [35]. Insertion of methylene group (**6i**) between the amide and 3-(trifluoromethyl)phenyl group of **6h,** and replacement of the amide with urea linkage (**6j**), as influenced by regorafenib and ripretinib, caused loss of enzymatic inhibitory activities.

To explore the effects of an additional substituent [36] on the 3-(trifluoromethyl)phenyl group of **6h**, we synthesized **6k**–**o** and evaluated enzymatic inhibitory activities. The results reveal that addition of 4-dimethylamino (**6k**), 4-morpholino (**6l**), and 4-methylpiperazino (**6m**) moiety on the *para*-position of the 3-(trifluoromethyl)phenyl group resulted in 2.3- to 5.6-fold enhanced activities (**6k**, IC_50_ = 4.31 μM; **6l**, IC_50_ = 1.76 μM; **6m**, IC_50_ = 2.17 μM) compared with **6h**. In contrast, 3-morpholino (**6n**) and 4-methylpiperazino (**6o**) on the *meta*-position of 3-(trifluoromethyl)phenyl group brought about lower enzymatic inhibitory activities (**6n**, IC_50_ = inactive; **6o**, IC_50_ = 5.03 μM) relative to **6l** and **6m**. Therefore, *para*-substitution led to higher activities than *meta*-substitution.

Further investigation of *para*-substituents of the 3-(trifluoromethyl)phenyl group revealed that 4-morpholinomethyl group (**6p**) and (1,1-dioxidothiomorpholino)methyl group (**6q**) caused lower activities relative to 4-morpholino group (**6l**). Gratifyingly, **6r** having 4-((4-methylpiperazin-1-yl)methyl)-3-(trifluoromethyl)phenyl group possessed 15.5-fold enhanced activity (IC_50_ = 0.14 μM) compared to **6m** (IC_50_ = 2.17 μM). The enzymatic inhibitory activity of **6r** was comparable to that of sunitinib (IC_50_ = 0.14 μM) and 1.9-fold higher than that of imatinib (IC_50_ = 0.27 μM). Encouraged by the promising enzymatic inhibitory activity of **6r**, further exploration of the 4-methylpiperazine group on **6r** was conducted. Removal of the 4-methyl group (**6s**) resulted in similar enzymatic inhibitory activity (IC_50_ = 0.37 μM) but substitution of the 4-methyl group with 2-hydroxyethyl group (**6t**) and acetyl group (**6u**) led to decreased activities. The derivatives with 4-(3-(dimethylamino)piperidin-1-yl)methyl group (**6v**) and 4-((2-(dimethylamino)ethyl)(methyl)amino)methyl group (**6w**) were less active than **6r**. On the basis of these results, the tail group of **6r** was selected and the R_2_ group was further optimized.

Next, we focused on SAR study for the R_2_ group. The effects of R_2_ groups were explored by replacing the primary amino group on **6r** with the amide groups (**7a**–**c**). The results show that derivatives possessing cyclohexyl amide (**7a**, IC_50_ = 1.51 μM) and benzamide (**7b**, IC_50_ = 0.74 μM) had 5.3- to 10.8-fold decreased activities relative to **6r** (IC_50_ = 0.14 μM). On the other hand, the activity of **7c** having the acetamide group (IC_50_ = 0.10 μM) was slightly higher than that of **6r**. Similarly, among the derivatives **6l**, **7d**, and **7e**, **7e** possessing the acetamide group had slightly improved activities (**6l**, IC_50_ = 1.76 μM; **7d**, IC_50_ = inactive; **7e**, IC_50_ = 0.88 μM). The additional derivatives having the acetamide group (**7f**–**h**) were more active compared with the derivatives possessing the 2-amino group (**6p**, **6v**, and **6w**). 

We assessed anti-proliferative activities of thiazolo[5,4-*b*]pyridine derivatives on c-KIT dependent cancer cells such as GIST-T1 and HMC1.2 (Table 2). The GIST-T1 was strongly positive for c-KIT and CD34, indicating that GIST-T1 cells had the characteristics of GIST [37]. The HMC1.2 cells bearing both c-KIT V560G in intracellular juxta-membrane region and c-KIT D816V in the kinase catalytic region caused SCF-independent constitutive activation of c-KIT [38,39]. The anti-proliferative activities of **6r**, **6s**, **7c**, and **7h** on GIST-T1 cancer cells were comparable or slightly higher relative to that of imatinib (GI_50_ = 0.02 μM). On HMC1.2 cells, the activities of **6k**, **6o**, **6r**, **6s**, and **7c** were higher than that of sunitinib (GI_50_ = 2.53 μM). It is worth noting that **6r** possessed 23.6-fold higher anti-proliferative activity (GI_50_ = 1.15 μM) on HMC1.2 cells than imatinib (GI_50_ = 27.10 μM). The SAR results reveal that **6r**, **6s**, and **7c** were able to strongly suppress the proliferation of GIST-T1 and HMC1.2 cancer cells.

Next, we evaluated enzymatic inhibitory activities of **6r**, **6s**, and **7c** against the c-KIT V560G/D816V double mutant (Table 3) [19]. These derivatives displayed 4.9- to 8.0-fold enhanced potency compared to imatinib (IC_50_ = 37.93 μM) and possessed comparable activities to sunitinib (IC_50_ = 3.98 μM) against the c-KIT V560G/D816V double mutant. Moreover, anti-proliferative activities of the **6r**, **6s**, and **7c** on c-KIT D816V Ba/F3 cells were evaluated (Table 4). On c-KIT D816V Ba/F3 cells, **6r** (GI_50_ = 1.11 μM), **6s** (GI_50_ = 1.06 μM), and **7c** (GI_50_ = 0.53 μM) had comparable activities to sunitinib (GI_50_ = 0.58 μM). It is worthwhile to note that **6r**, **6s**, and **7c** possessed higher differential cytotoxicity than sunitinib on c-KIT D816V Ba/F3 cells relative to parental Ba/F3 cells, which implies that **6r**, **6s**, and **7c** would be more selective c-KIT inhibitors relative to sunitinib. These efforts led to the identification of novel thiazolo[5,4-*b*]pyridine derivatives that possessed enhanced anti-proliferative activities on c-KIT D816V Ba/F3 cells.

### 3.3. Molecular Docking Studies of **6j**, **6r**, and **7c** with c-KIT

To investigate the binding mode of thiazolo[5,4-*b*]pyridine derivatives on c-KIT, we performed molecular docking studies of **6j**, **6r** and **7c** using the X-ray co-crystal structure of c-KIT complexed with imatinib (Figure 2). For comparison, the binding mode of imatinib on c-KIT is presented in Figure 2C (PDB code: 1T46) [40]. The results shows that **6r** and **7c** form a pair of hydrogen bonding with a Cys673 backbone in the hinge region. In addition, **6r** and **7c** also makes hydrogen bonding with Glu640/Asp810 and Ile789/His790 backbone, and these hydrogen bondings are also observed in the co-crystal structure of imatinib and c-KIT. Moreover, the thiazolo[5,4-*b*]pyridine moiety of **6r** and **7c** participates in hydrophobic interactions with Leu799, Val603, Ala621, and Val654. Furthermore, the 3-trifluoromethyl group of **6r** and **7c** fits well into the hydrophobic binding pocket which is also occupied by ponatinib [35]. However, **6j** could not form a hydrogen bonding with Asp810 observed in imatinib and could not fit into the hydrophobic binding pocket occupied by ponatinib. The predicted binding model suggests that **6r** and **7c** possess inhibitory activity, and **6j** is inactive against c-KIT.

### 3.4. Kinase Panel Profiling of **6r**

Kinase panel profiling of **6r** showed that 13 of 371 kinases (c-KIT, RAF1, DDR1, RET, FMS, FRK, ARAF, LYN, BRAF, LCK, PDGFRa, P38a, PDGFRb) were inhibited more than 90% by **6r** (1 μM). The results reveal that **6r** possesses reasonable kinase selectivity (Figure 3 and Appendix A).

### 3.5. Inhibition of c-KIT Signaling in GIST-T1, HMC1.2, and c-KIT D816V Ba/F3 Cells

To assess that **6r**, **6s**, and **7c** are capable of inhibiting c-KIT auto-phosphorylation and suppressing downstream signaling, Western blot analysis was assessed using GIST-T1, HMC1.2, and c-KIT D816V Ba/F3 cells. Compound **6r**, **6s**, **7c**, and imatinib remarkably inhibited the levels of phosphorylated c-KIT (Tyr703) and its downstream signaling mediators including p-AKT, p-ERK, p-PLCγ and p-STAT3 in GIST-T1 cells (Figure 4A,B and Appendix A). On HMC1.2 and c-KIT D816V Ba/F3 cells, **6r**, **6s**, and **7c** at 10 μM concentration completely inhibited c-KIT auto-phosphorylation but not imatinib (Figure 4C–F and Appendix A). These results show that **6r**, **6s**, and **7c** blocked c-KIT downstream signaling associated with their anti-proliferation activities on GIST-T1, HMC1.2, and c-KIT D816V Ba/F3 cells.

### 3.6. Induction of Apoptosis and Cell Cycle Arrest in GIST- T1 and HMC1.2 Cells

Apoptosis and cell cycle arrest are important in cancer cell death [42,43]. PI3K/mTOR signaling by mutant c-KIT plays an important role in the tumor cell proliferation and apoptosis in GIST [44,45]. Moreover, c-KIT inhibitor induces apoptosis against mast cells [46]. The effects of **6r**, **6s**, and **7c** on apoptosis induction and cell cycle arrest were assessed in GIST-T1, HMC1.2, and c-KIT D816V Ba/F3 cells. As a result, **6r**, **6s**, and **7c** remarkably induced production of apoptotic cells in GIST-T1 (Figure 5A,B), HMC1.2 (Figure 5E,F), and c-KIT D816V Ba/F3 (Figure 5I,J) cells. It is worth noting that treatment with imatinib (1 μM) for 48 h did not induce apoptosis in HMC1.2 cells (Figure 5E,F). Moreover, induction of apoptosis by **6r**, **6s**, and **7c** was measured using apoptosis markers, which displayed that cleavage of PARP was induced in HMC1.2 cells (Figure 5M and Appendix A). We performed FACS analysis to determine whether **6r**, **6s**, **7c**, and imatinib induce cell cycle arrest in GIST-T1 and HMC1.2 cells. Treatment with **6r**, **6s**, **7c**, and imatinib (0.01 μM) induced G0/G1 arrest in 25–35% compared to control in GIST-T1 cells (Figure 5C,D). In addition, treatment with **6r**, **6s**, and **7c** (0.5 μM) for 24 h induced G0/G1 arrest more strongly in HMC1.2 and c-KIT D816V Ba/F3 cells compared with imatinib (Figure 5G,H,K,L). Collectively, these results indicate that **6r**, **6s**, and **7c** inhibit proliferation of GIST-T1, HMC1.2, and c-KIT D816V Ba/F3 cells through induction of apoptosis and cell cycle arrest.

### 3.7. Suppression of Migration and Invasion in GIST-T1 Cells

c-KIT is related to the migration and invasion of GIST cells [47]. To identify the effects of **6r**, **6s**, and **7c** on migration, we carried out a wound healing assay in the GIST-T1 cells (Figure 6A). Cell migration treated with **6r**, **6s**, **7c**, and imatinib (0.005 μM and 0.05 μM) was decreased by 50–70% compared to that of control (Figure 6C). Moreover, the migration ratio of GIST-T1 cells treated with **6r**, **6s**, **7c**, and imatinib deceased with increasing concentration. We conducted an invasion assay to further test the effects of **6r**, **6s**, and **7c** on invasion (Figure 6B) and found that **6r**, **6s**, and **7c** were capable of significantly suppressing the invasion capability of GIST-T1 cells (Figure 6D).

### 3.8. Suppression of Anchorage-Independent Growth in GIST-T1 Cells

Pharmacological inhibition and knockdown of c-kit decreased the colony growth of GIST cells [48]. A soft agar assay in GIST-T1 cells was performed to evaluate whether **6r** blocks anchorage-independent growth (Figure 7). Treatment with two different concentrations of **6r** and imatinib for 3 weeks resulted in dramatic inhibition of anchorage-independent growth of the GIST-T1 cells. Notably, incubation of **6r** at 0.05 μM completely suppressed anchorage-independent growth compared to control, indicating that blockade of c-KIT by **6r** suppresses tumorigenesis of GIST-T1 cells.

## 4. Conclusions

Dysregulation of c-KIT is associated with GIST. In order to identify a unique class of c-KIT inhibitors capable of inhibiting clinically relevant c-KIT mutations and overriding imatinib resistance, we designed and synthesized 31 novel thiazolo[5,4-*b*]pyridine derivatives and carried out structure-activity relationship studies against c-KIT enzyme and c-KIT activated cells. The results of SAR studies show that **6r**, **6s**, and **7c** possessed higher enzymatic and cellular activities than imatinib and these three derivatives were comparable to sunitinib in terms of activities. In particular, **6r** hand a 8.0-fold higher enzymatic inhibitory activity against c-KIT V560G/D816V double mutant and 23.6-fold higher anti-proliferative activity on HMC1.2 cells than imatinib, indicating that **6r** would be a promising lead for overriding imatinib resistance. It is of note that, compared with sunitinib, **6r** had a higher differential cytotoxicity effect on c-KIT D816V Ba/F3 cells relative to parental Ba/F3 cells, which suggests that **6r** is more selective c-KIT D816V inhibitor than sunitinib.

The analysis of molecular docking studies suggests that **6r** forms hydrogen bonding with a Cys673 backbone, Ile789/His790 backbone, and Glu640/Asp810. Moreover, the 3-trifluoromethyl group of **6r** occupies the hydrophobic binding pocket formed by Leu647, Ile653, Leu783, and Ile808. Kinase panel profiling showed that **6r** possesses a reasonable kinase selectivity and potent activities against 13 out of 371 kinases including c-KIT.

Western blot analysis revealed that **6r** significantly blocks c-KIT auto-phosphorylation and downstream signaling in GIST-T1, HMC1.2, and c-KIT D816V Ba/F3 cells. Compounds **6r**, **6s**, and **7c** induce more significantly apoptosis than imatinib on HMC1.2 cells. In addition, **6r**, **6s**, and **7c** are capable of blocking cell motility and suppressing anchorage-independent growth of GIST-T1 cells. Taken together, the novel thiazolo[5,4-*b*]pyridine derivatives provide insights into the design of novel c-KIT inhibitors capable of overcoming imatinib resistance.

## Data Availability

Not applicable.

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
