# Peer review of "Identification of Thiazolo[5,4-b]pyridine Derivatives as c-KIT Inhibitors for Overcoming Imatinib Resistance"

_cancers, 2022, doi:10.3390/cancers15010143_

Round 1

Reviewer 1 Report

The authors described a novel compound that have strong inhibitory effect on Imatinib-resistance GIST cells. Although similar derivative has been found to have inhibitory effect on PI3K, this is the first report of such compound on c-KIT. In addition, the mechanisms and selectivity have also been explored. The English is excellent and the writing is error free. The findings are significant and have potential to be used clinically. I recommend the article to be published in present form.

Author Response

I am deeply grateful to you for your kind consideration and useful directions. All comments on my manuscript are very helpful to rectify inappropriate descriptions and boost the quality of my manuscript. We appreciate you and all reviewers. Based on the reviewers' comments, the original manuscript has been revised. Included are point-by-point responses to all the comments from the reviewers. I hope that these responses would be suitable for all the comments.

Reviewer 2 Report

The manuscript entitled „Identification of Thiazolo[5,4-b]pyridine Derivatives as c-KIT Inhibitors for Overcoming Imatinib Resistance” by Nam et al. describes fully described synthesis of 31 novel derivatives as the potential c-KIT inhibitors lacking the issues of imatinib (like drug resistance). In general, the manuscript provides useful novel data in the topic field, however, I see several shortcomings needed to be addressed by the authors before the publication.

-          The introduction is written well, but in my opinion does not provide a necessary background for the reader, especially in the area of compound novelty. I understand that the synthesized compounds are new (I checked several and didn’t find any previous report about them), but clearly other attempts to use thiazolo[5,4-b]pyridine derivatives were undertaken (not necessarily c-KIT but as kinases inhibitors for sure). I would expect that the authors will provide background about thiazolo[5,4-b]pyridine derivatives' potential as kinases inhibitors already described in the literature, but also about other (structurally not related) derivatives designed ans synthesized to address the issue of imatinib resistance. At the same time paragraph line 60-75 is rather unnecessary for me since the authors mix their compounds in proof-of-concept/early pre-clinical studies with clinically-approved drugs.

https://www.ncbi.nlm.nih.gov/pmc/articles/PMC7594053/

https://onlinelibrary.wiley.com/doi/full/10.1002/jhet.3697

- The number of significant digits, especially in in cellulo studies, is too high, CellTiter-Glo assay just cannot provide a resolution so high to assess GI50 (which is a parameter derived and extrapolated from crude data) with so many significant digits. It gives a false image of assay sensitivity/resolution

-          The methodology for kinase panel profiling is missing; Table S1 title (In vitro selectivity profiling of 6r at 1.0 μM against 371 human kinases) suggests that the assay was done using all of those kinases. Is my assumption correct? That would be highly expensive and rather unnecessary at that stage of the studies.

-          Western blot analyses miss a necessary control – total (unphosphorylated) protein levels. It is a standard approach that densitometry for phosphorylated protein is related to the total level of the protein. Without such control reader (as well as the authors) are unable to say if the e.g. p-c-KIT level dropped because of the phosphor/de-phospho ratio switch or because of the significant decrease in total protein level. Such WB should be provided at least for c-KIT.

-          Statistical analysis of the data is done wrong. First, the authors do not provide information about the number of samples/repeated measurements. Secondly, the authors do not provide information about specific tests used during statistical analyses (one-way ANOVA? what post-test was used). Thirdly, p<0,001 (even p<0,01 is too much) is definitely an overkill here. I assume that each test was performed in three independent repeats (am I right?). In that case, it is even not possible to assess if the data have a normal distribution (some tests will even not accept datasets with 3 samples to do it). P<0,05 is the highest significance that should be reported for this type of in vitro study (no matter what GPP says), and should always be accompanied by N, the test used, and the reference to which group the significance was tested (I assume to vehicle-treated samples). In some cases, it would be better to analyze the difference between the compounds and imatinib (at the same concentration).

-          Lack of any discussion. In section 3 authors describe their results (and they are doing it pretty well), but without any reference to the literature. Along with a lack of a proper paragraph in the introduction (mentioned above), it suggests that the manuscript creates a completely new area of study, never addressed by other scientists (which is obviously not true, both imatinib resistance and thiazole[5,4-b]pyridine derivatives biological potential is a widely investigated topic.

-          Why cell cycle analysis on the c-KIT D816V Ba/F3 cell line was omitted? Clearly, the results from anti-proliferative and WB assays pointed out that HMC1.2 CL provides the most profound differences between selected compounds and imatinib, but cell cycle analysis on c-KIT D816V Ba/F3 CL is missing for a full image.

-          Results from cell cycle analysis and apoptosis rate indicate that the most significant differences between imatinib and tested compounds occur on apoptosis level, but not on cell cycle – can authors discuss this issue shortly – could it be derivative specific?

Minor issues: some typos (line 17, missing/too many spaces in lines 102-108, line 220)

Author Response

(The authors gave the same response as above.)

Reviewer 3 Report

Drs. Nam et al have developed novel thiazole[5,4-b]pyridine compounds, using a structure-activity relationship (SAR) approach, in order to identify novel inhibitors of imatinib-resistant, mutant c-KIT for the treatment of gastrointestinal stromal tumors (GIST).  The authors synthesized a small series of 31 compounds and tested them against GIST cell lines containing either wt c-KIT or imatinib-resistant lines harboring mutant c-KIT and a double-mutant c-KIT.  The authors evaluated the activities of the compounds for their kinase inhibitory activities and effects on c-KIT signaling.  They also studied the effects of these compounds in a series of cellular assays (growth inhibition, apoptosis, cell cycle activity, anchororage-indepent growth, migration, invasion).  Also included was a molecular docking study.  Overall, the authors identified three compounds, especially one (6r), that were as good or better than imatinab, especially in the imatinab-resistant enzymes and cells.  Overall, this is a straightforward study that may offer novel compounds to treat imatinab-resistant GIST.  However, for clarity, the authors should address the following:

1.      Lines 103 – 106 and 237 - 240: Please provide references for the cell lines and references or the authors’ own independent verification that the presumptive mutant c-KIT in the cell lines was indeed mutant.

2.      Line 191: “Enzymatic activities” of all synthesized … is confusing.  The sentence should state: “Enzymatic inhibitory activities” of all synthesized…. .

3.      Line 257: The use of “(Table 3)” is confusing.  Are the authors referring to the antiproliferative activities in Table 2?  Or to the enzymatic inhibitory activities and anti-proliferative activites in Table 3?

4.      In regard to #3, above, Table 3 can be made more clear and have “enzymatic inhibitory activities” above or next to the column headed by “IC50” and have anti-proliferative activities above or next to the two columns headed by “Ba/F3 cells…”

5.      Section 3.3 – Molecular docking with 6r compared to imatinib.  Why did the authors not also test 7c and one of the “inactive” compounds to get a better sense as to why these compounds were inactive?

6. Related, can these compounds be competed out by ATP?  Can 6r compete imatinib for binding, enzymatic activity and the like?  In other words, what is the basis of the binding of 6r and other compounds to c-KIT to the ATP pocket?

7. Minor but important:

a.       Line 106; 10 ng/mL IL-3 is impossible!  It should be: 10 ng IL-3/mL.  Similarly, line 141 should read: 50 ug propidium iodide/mL and line 142 should read: 1 mg RNase A/mL (not 1 mg/mL RNase A!)

b.      Line 355: should read: Dysregulation of c-KIT is associated with GIST.”

 Overall, this is a well-done study that may eventually provide novel agents to inhibit mut c-KIT in GIST.  The authors should address the concerns for clarity.

Author Response

(The authors gave the same response as above.)

Round 2

Reviewer 2 Report

I find most of the concerns as adequately addressed. I still treat the lack of total c-KIT analysis at the western blot stage as an important flaw, but I leave the decision of how to address it to the editor.